# Approaches to quantify the contribution of multiple anemia risk factors in children and women from cross-sectional national surveys

Yi-An Ko[1]*, Anne M. Williams[2], Janet M. Peerson[3], Hanqi Luo[4], Rafael Flores-Ayala[5], James P. Wirth[6], Reina Engle-Stone[3], Melissa F. Young[4], Parminder S. Suchdev[4,5,7]

**1** Department of Biostatistics and Bioinformatics, Rollins School of Public Health, Emory University, Atlanta, Georgia, United States of America, **2** Department of Human Nutrition, University of Otago, Dunedin, Otago, New Zealand, **3** Department of Nutrition and Institute for Global Nutrition, University of California, Davis, California, United States of America, **4** Department of Global Health, Rollins School of Public Health, Emory University, Atlanta, Georgia, United States of America, **5** Nutrition Branch, Centers for Disease Control and Prevention (CDC), Atlanta, Georgia, United States of America, **6** GroundWork, Fläsch, Switzerland, **7** Department of Pediatrics, Emory School of Medicine, Emory University, Atlanta, Georgia, United States of America

* yi-an.ko@emory.edu

**Data Availability Statement:** The original data from this study are provided by individual countries

## Abstract

### Background

Attributable fractions (AF) of anemia are often used to understand the multifactorial etiologies of anemia, despite challenges interpreting them in cross-sectional studies. We aimed to compare different statistical approaches for estimating AF for anemia due to inflammation, malaria, and micronutrient deficiencies including iron, vitamin A, vitamin B12, and folate.

### Methods

AF were calculated using nationally representative survey data among preschool children (10 countries, total N = 7,973) and nonpregnant women of reproductive age (11 countries, total N = 15,141) from the Biomarkers Reflecting Inflammation and Nutrition Determinants of Anemia (BRINDA) project. We used the following strategies to calculate AF: 1) Levin's formula with prevalence ratio (PR) in place of relative risk (RR), 2) Levin's formula with odds ratio (OR) in place of RR, and 3) average (sequential) AF considering all possible removal sequences of risk factors. PR was obtained by 1) modified Poisson regression with robust variance estimation, 2) Kleinman-Norton's approach, and 3) estimation from OR using Zhang-Yu's approach. Survey weighted country-specific analysis was performed with and without adjustment for age, sex, socioeconomic status, and other risk factors.

### Results

About 20–70% of children and 20–50% of women suffered from anemia, depending on the survey. Using OR yielded the highest and potentially biased AF, in some cases double those using PR. Adjusted AF using different PR estimations (Poisson regression, Kleinman-

through the BRINDA project, and most of the country survey data are not publicly available. Data described in the manuscript can be made available upon request pending approval from the BRINDA steering committee and country representatives. Should researchers like to collaborate or request access to this data, please connect through https://www.brinda-nutrition.org/.

**Funding:** This study was supported by the Bill & Melinda Gates Foundation in the form of a grant (0000046472) awarded to MFY, PSS, JMP, HL, & YAK. This study was also supported by the Centers for Disease Control and Prevention in the form of grants: (19IPA1905154) awarded to PSS, (22IPA2210040) awarded to MFY, and (0000066978) awarded to MFY, PSS, JMP, HL, & YAK. This study was also supported by Eunice Kennedy Shriver National Institute of Child Health and Human Development, HarvestPlus, and the United States Agency for International Development. The funders had no role in study design, data collection and analysis, decision to publish, or preparation of the manuscript.

**Competing interests:** The authors have read the journal's policy and have the following competing interests: JPW is an employee of GroundWork LLC. This does not alter our adherence to PLOS ONE policies on sharing data and materials. There are no patents, products in development or marketed products associated with this research to declare.

Norton, Zhang-Yu) were nearly identical. Average AF estimates were similar to those using Levin's formula with PR. Estimated anemia AF for children and women were 2–36% and 3–46% for iron deficiency, <24% and <12% for inflammation, and 2–36% and 1–16% for malaria. Unadjusted AF substantially differed from adjusted AF in most countries.

## Conclusion

AF of anemia can be estimated from survey data using Levin's formula or average AF. While different approaches exist to estimate adjusted PR, Poisson regression is likely the easiest to implement. AF are a useful metric to prioritize interventions to reduce anemia prevalence, and the similarity across methods provides researchers flexibility in selecting AF approaches.

## Introduction

Anemia remains a major global health problem that contributes to increased morbidity and mortality, particularly in women and children [1, 2] Global goals to reduce the prevalence of anemia in women of reproductive age by 50% by the year 2025 are not on track, and the prevalence of anemia in children remains high [3], despite concerted efforts by global- and country-level stakeholders to implement interventions (e.g., iron supplementation, micronutrient fortification, malaria reduction) in areas with a high prevalence of anemia. Additionally, the USAID Advancing Nutrition Anemia Task Force [4] has underscored the importance of identifying the underlying biological mechanisms of anemia and its assessment methodologies. Thus, it is imperative to better understand the overlapping role of modifiable and non-modifiable causes of anemia, to determine what interventions can have the greatest ability to reduce the anemia prevalence in specific population groups.

Common causes of anemia include nutritional deficiencies, infectious and inflammatory conditions, blood loss, and genetic hemoglobin disorders, which may overlap and vary based on geography [1, 5, 6]. For decades, iron deficiency was thought to be the primary cause of anemia [7], but many studies have demonstrated the proportion of anemia attributable to iron deficiency to be lower than 50%, especially in settings with high infection burden [5, 6, 8–10]. Most public health guidelines acknowledge the multifactorial etiology of anemia, but in both clinical and public health practice, iron is often the first and often only intervention to address anemia [11]. If anemia resolves, diagnosis of iron deficiency anemia is presumed. Efforts are needed to understand the contribution of diverse factors to anemia.

Quantifying the contribution of multiple anemia risk factors that often overlap with one another can be complex. The attributable fraction (AF) in a population is an epidemiologic measure often used to quantify the proportional reduction in population risk of disease or mortality over a specified time interval resulting from eliminating the exposure(s) of interest while distributions of other risk factors remain unchanged in the population [12, 13]. The term "attributable" has a causal interpretation; that is, AF is the estimated fraction of all cases that would not have occurred if the exposure had been removed. As such, AF is meaningful when the exposure is causally related to the disease outcome and when the exposure is amenable to intervention programs [14]. As many diseases are caused by multiple exposures of risk factors simultaneously, and individual exposures may interact in their impact on disease, AFs

for individual exposures might add up to more (or less) than the AF for all the exposures and even more than 100% [15].

A number of methods to calculate AF have been established and reviewed, especially in cross-sectional studies and cohort studies where sampling is independent of exposure status and confounding may exist [16–18]. Several prior surveys have tried to estimate AF of anemia due to malaria, inflammation or infection, and deficiency of iron or other micronutrients in individual countries [5, 8–10, 19–22]. However, different equations have been used to calculate AF, and various methods have been used to estimate the adjusted risk or prevalence ratios using Poisson regression or odds ratios using logistic regression. As such, comparison of AF of anemia among different risk factors and across different studies can be challenging. Previously, Rückinger et al. compared different approaches to estimate AF for established risk factors of cardiovascular disease using the National Health and Nutrition Examination Survey (NHANES) and found that AF following the Levin's approach (using odds ratios from logistic regression) yielded higher estimates than others [18]. Regardless, studies that compare different approaches [12] to quantify AF of anemia using data from multiple national surveys are limited. Given that different estimates of AF may impact programmatic decisions, it is important to understand the extent to which different AF methods yield comparable results.

The objective of the study was to compare the use of different AF approaches to estimate relative contributions of multiple known risk factors for anemia using national survey data. Our goal was not to define specific AF values to be used by anemia control programs, but to assess the pros and cons of different approaches for AF estimation using pooled nutrition survey data from various geographies. Findings will help researchers and public health practitioners to select appropriate methods to quantify the scope and strength of individual risk factors for anemia in order to set intervention priorities.

## Methods

### Study population and analysis sample

This was a secondary data analysis based on nationally representative survey data including preschool children (PSC, aged 6–59 months) and non-pregnant women of reproductive age (WRA, aged 15–49 years) from the BRINDA (Biomarkers Reflecting Inflammation and Nutrition Determinants of Anemia) project (brinda-nutrition.org). Details of the BRINDA project objectives and methodology have been described previously [23]. The BRINDA protocol was reviewed by the Institutional Review Board of the NIH and was deemed non–human subjects research. Data eligible for inclusion in the BRINDA project contain at least hemoglobin and ≥1 biomarker for acute inflammation, either C-reactive protein (CRP) or α-1-acid glycoprotein (AGP) [24]. The inclusion criteria for the current study were: availability of hemoglobin, ferritin, both markers for inflammation (CRP and AGP), age, sex, and household socioeconomic status (SES). This resulted in an analysis data set representing PSC in 10 countries and WRA in 11 countries (year indicating the last year of the survey), including Côte d'Ivoire (2007), Cameroon (2009), Liberia (2011), Malawi (2016), Azerbaijan (2013), Cambodia (2014), Pakistan (2011, WRA only), Afghanistan (2013), Bangladesh (2012), Laos (2006), and Nepal (2016).

### Anemia and exposure variables

This study was designed to estimate the AFs for anemia with respect to factors with an established biological role in anemia that are generally amenable to public health interventions, based on a previously published anemia conceptual framework from the BRINDA project [1, 5, 6, 23]. Only proximal and individual exposures or risk factors of anemia were considered

for AF estimation. These exposure variables included micronutrient deficiencies (iron, vitamin A, vitamin B12, and folate), inflammation, and malaria. Other potential covariates associated with anemia in some settings (e.g., zinc deficiency, BMI, stunting, wasting) were not included given prior analyses from the BRINDA group that showed inconsistent associations with anemia [6, 25]. Additionally, stunting and wasting in children may not be direct causes of anemia but rather a consequence of inflammation or micronutrient deficiencies [1]. Hemoglobin concentrations were measured by HemoCue (201+ or 301) in all surveys and were adjusted for altitude and smoking status (for women only) following World Health Organization (WHO) guidelines [26]. Adjusted hemoglobin <110 g/L and <120 g/L were used to define anemia in children and non-pregnant women, respectively. Inflammation was defined as either CRP >5mg/L or AGP >1 g/dL [27]. Inflammation-adjusted (using the BRINDA regression adjustment approach [28]) ferritin concentrations <12 µg/L and <15 µg/L were used to define iron deficiency (ID) in children and women, respectively [29]. Vitamin A deficiency (VAD) was defined as inflammation-adjusted retinol binding protein (RBP) or retinol <0.7 µmol/L for PSC. Since VAD is rare in women, we defined vitamin A insufficiency as RBP or retinol <1.05 µmol/L for WRA [30, 31] These regression adjustments for inflammation used survey-specific coefficients and previously established pooled deciles for CRP and AGP [28, 31, 32]. Folate deficiency was defined as red blood cell folate <226.5 nmol/L or serum folate <10 nmol/L for radioimmunoassay (<6.8 nmol/L for microbiological assay) [33]. Vitamin B12 deficiency was defined as serum or plasma B12 <150 pmol/L [34]. Malaria infection was defined as either positive or negative based on blood smears (Cote d'Ivoire) or rapid diagnostic test (Cameroon, Liberia and Malawi). In addition, we estimated the AF related to blood disorders for PSC in Malawi, Cambodia, and Nepal. Any blood disorder was defined as the presence of sickle cell trait or disease, HbE-heterozygote or HbE-homozygote, or α-Thalassaemia or β-Thalassaemia trait or disease. Given the cross-sectional survey design in all countries, both anemia status and predictors were assessed at the same time.

## AF calculations

The traditional and most frequently applied approach for estimating AF is Levin's formula [35]. Assuming equal proportions of expected cases among the exposed and unexposed and that excess cases among the exposed are attributable to the risk factor, Levin's approach only requires the prevalence of the exposure/risk factor ($p_e$) and the relative risk or risk ratio (RR),

$$AF_{Levin} = \frac{p_e(RR - 1)}{p_e(RR - 1) + 1}.$$ (1)

Since $RR = P_1/P_0$, where $P_1$ and $P_0$ indicate the probabilities or risks of having disease in the exposed and unexposed groups, respectively. Eq (1) can also be expressed as:

$$AF_{Levin} = \frac{p_e(P_1 - P_0)}{p_e(P_1 - P_0) + P_0}.$$ (2)

As demonstrated by Bruzzi et al. and others, variations of Levin's formula to provide AF adjusted for other known risk factors include: using adjusted RR and replacing RR with adjusted odds ratio estimates (OR) from logistic regression [36–38]. The adjusted AF quantifies the proportion of the diseased population that would be prevented if the risks of disease in the exposed population were changed to the risks of the unexposed population, leaving the rest of adjusted variables unchanged. Rockhill recommended the following alternative expression

[12]

$$AF_{Adj} = pd \frac{(\widehat{RR} - 1)}{\widehat{RR}}, \tag{3}$$

where $pd$ is "the proportion of total cases in the population arising from the exposure category" or proportion of cases who are exposed to the risk factor. Note that

$$pd = \frac{p_e \widehat{P}_1}{p_e \widehat{P}_1 + (1 - p_e)\widehat{P}_0} = \frac{p_e \widehat{P}_1}{p_e(\widehat{P}_1 - \widehat{P}_0) + \widehat{P}_0},$$

hence, Eq (3) is a re-expression of Levin's formula.

It is important to note that RR in Eq (1) can be approximated by OR when the disease is rare in the study population. However, when the disease is not rare, the OR cannot approximate RR. The more frequent the disease, the more the OR overestimates or underestimates the RR. Since anemia is common among women and young children in low- and middle-income countries, using OR is not appropriate. In the following applications, we demonstrate the use of OR to directly calculate AF for illustrative purposes.

When the disease condition is common, Zhang and Yu [39] proposed to convert the adjusted OR obtained from logistic regression to approximate RR by

$$\widehat{RR} = \frac{OR}{(1 - \widehat{P}_0) + \widehat{P}_0 \times OR} \tag{4}$$

where $\widehat{P}_0$ indicates the crude risk estimate of having the disease in the unexposed group.

Kleinman and Norton [40] proposed to derive the adjusted RR using the ratio of the average predicted risk conditional on all observations being exposed, to the average predicted risk conditional on all observations being unexposed. Individual predicted probabilities of the disease are estimated by logistic regression as if the individual had or had not been exposed. Suppose the overall sample size is $N$, the estimated risk or probability of having the disease for individual $i$ is denoted as $p_i$, and the covariates for individual $i$ are denoted as $X_i$, then the adjusted RR can be estimated by:

$$\widehat{RR} = \frac{\frac{1}{N}\sum_{i=1}^{N} p_i \left(X_i | as\ if\ exposed\right)}{\frac{1}{N}\sum_{i=1}^{N} p_i \left(X_i | as\ if\ unexposed\right)} = \frac{\widehat{P}_1^*}{\widehat{P}_0^*}. \tag{5}$$

This $\widehat{RR}$ can then be used in Eq (1) to estimate AF.

An alternative adjusted AF can be calculated following similar steps in the Kleinman and Norton's approach [17, 18, 41] That is, fit a logistic regression model containing all known risk factors of disease, including the exposure variable of interest. Next, classify all individuals as unexposed and estimate the probability of being a case for each individual using the re-classified data. Finally, compute the adjusted AF based on the proportion of disease cases that can be attributed to exposure as:

$$AF_{Adj} = \frac{(Observed\ number\ of\ cases - Expected\ number\ of\ cases)}{Obseved\ number\ of\ cases}, \tag{6}$$

where the "expected number of cases" is the sum of predicted probabilities $\sum_{i=1}^{N} p_i \left(X_i | as\ if\ unexposed\right)$ in Eq (5). Since the observed number of cases in the data sample is $N \times (p_e(\widehat{P}_1 - \widehat{P}_0) + \widehat{P}_0)$ and the expected number of cases is $N \times \widehat{P}_0^*$, Eq (6) can be

expressed as:

$$AF_{Adj} = \frac{p_e(\widehat{P}_1 - \widehat{P}_0) + \widehat{P}_0 - \widehat{P}_0^*}{p_e(\widehat{P}_1 - \widehat{P}_0) + \widehat{P}_0} = \frac{p_e(\widehat{RR} - 1) + (1 - \frac{\widehat{P}_0^*}{P_0})}{p_e(\widehat{RR} - 1) + 1}.$$

Again $\widehat{P}_1$ and $\widehat{P}_0$ indicate the crude risk estimates of having the disease in the exposed and unexposed groups, respectively. When $\widehat{P}_1 \approx \widehat{P}_1^*$ and $\widehat{P}_0 \approx \widehat{P}_0^*$, the results using Kleinman and Norton's approach to estimate RR in Levin's formula are very close to using Eq (6).

Eide and Gefeller [41] proposed the concepts of sequential and average AF to study the effect of stepwise removal of one exposure at a time in a pre-specified order. Sequential AF is defined as the "additional" effect by removing an exposure from the population when a previous set of exposures have been removed in the specified sequence based on Eq (6). For example, a sequential AF can be obtained by subtracting the adjusted AF with only the first exposure removed from the adjusted AF with both the first and second exposures removed. Note that sequential AF is different from AF in Eqs (1)–(3) and (6) and should not have the same interpretation. A single exposure can have multiple sequential AF values, depending on which other exposures are removed first. Although it may seem arbitrary to let the sequence influence the estimate, this issue can be resolved by using average AF. The average AF for an exposure is the average of all sequential AFs for this exposure over all possible removal orderings [41]. When there are a total of $m$ exposures of interest or explanatory variables, the average AF is an average of $m$! sequential AF values. Compared to sequential AF, average AF is independent to exposure removal ordering and yields an adjusted AF to one single exposure. Additional demographic covariates can be adjusted for during the step of fitting the logistic regression model. Rückinger et al. has published a SAS macro to calculate average AF with binary covariates in a logistic regression model [18] We modified the macro to incorporate complex survey design for our applications. It is worth noting that the availability of individual data is a requirement of average AF computation, and meta-analysis [10] based on summary or aggregated data (e.g., adjusted RR) is not feasible.

## Statistical analysis

All analyses accounted for complex survey design effects using cluster, strata and sampling weights. In Cameroon 2009 (PSC and WRA), Cambodia 2014 (PSC and WRA), Azerbaijan 2013 WRA and Côte d'Ivoire 2007 WRA survey data, hemoglobin, vitamin B12, or folate were measured in randomly selected smaller sub-samples. Thus, the sample sizes with all variables of interest consisted of 50–60% of the original samples. For the remaining surveys, the sample sizes of complete cases consisted of nearly 90% or more of the original samples (Tables 1 and 2). Given that the goal was to compare different AF calculation approaches, complete case analysis was used for all surveys. As such, sampling weights were re-calculated using those provided in the original country surveys to account for a smaller analytical sample size in order to produce nationally represented estimates.

Given the cross-sectional survey design, prevalence ratios (PR) were estimated (in place of RR in the Levin's AF formula). However, it is important to note that the prevalence of anemia is influenced by both incidence and duration, thus PR does not necessarily provide a valid estimate of RR. To account for clustering, modified Poisson regression that combines a Poisson regression model with robust variance estimation by generalized estimating equations was used [42]. Similarly, logistic regression with robust variance estimation was used to estimate OR. In the adjusted AF approach, models were adjusted for all exposures or risk factors of interest available in the survey dataset and distal-level risk factors of anemia, including age,

**Table 1. Prevalence of anemia and selected anemia risk factors in preschool children (PSC).**

| Country | Original Survey N | Analysis N | Anemia | Inflammation | Iron Deficiency | Vitamin A Deficiency | Vitamin B12 Deficiency | Folate Deficiency | Malaria | Blood Disorder |
|---|---|---|---|---|---|---|---|---|---|---|
| | | | | | *Malarial* | | | | | |
| **Côte d'Ivoire** | 744 | 728 | 71.0% | 67.8% | 39.3% | 2.8% | | | 27.3% | |
| **Cameroon** | 742 | 339 | 51.8% | 48.5% | 34.5% | 10.3% | 16.3% | 9.1% | 26.8% | |
| **Liberia** | 1434 | 1433 | 59.5% | 59.1% | 51.0% | 5.3% | | | 29.4% | |
| **Malawi** | 1100 | 972 | 30.1% | 55.7% | 21.4% | 8.1% | 4.7% | 0.2% | 26.9% | 47.4% |
| | | | | | *Non-malarial high inflammation* | | | | | |
| **Azerbaijan** | 1051 | 1050 | 23.9% | 31.0% | 22.3% | 6.6% | | | | 72.6% |
| **Cambodia** | 665 | 319 | 51.0% | 41.9% | 4.5% | 4.6% | 2.2% | 7.2% | | 72.6% |
| **Laos** | 481 | 478 | 40.5% | 43.9% | 25.6% | | | | | |
| | | | | | *Medium-to-low inflammation* | | | | | |
| **Afghanistan** | 660 | 595 | 40.0% | 23.9% | 24.2% | 39.0% | | | | |
| **Bangladesh** | 467 | 435 | 32.9% | 29.6% | 13.4% | 4.8% | | | | |
| **Nepal** | 1651 | 1624 | 19.2% | 27.9% | 26.5% | 6.4% | | 0.9% | 0.0% | 7.9% |

Anemia defined as hemoglobin adjusted for altitude <11.0 g/dL, inflammation defined as either CRP >5 mg/L or AGP >1 g/dL, iron deficiency defined as inflammation-adjusted ferritin <12 μg/L, vitamin A deficiency defined as inflammation-adjusted retinol binding protein or retinol <0.7 μmol/L, folate deficiency: red blood cell folate <226.5 nmol/L or serum folate <10 nmol/L for radioimmunoassay (<6.8 nmol/L for microbiological assay), vitamin B12 deficiency: serum or plasma B12 <150 pmol/L. Inflammation prevalence of >30% and ≤30% were used to define high inflammation and medium-to-low inflammation.

sex, and SES (defined below). In most countries, separate variables of various household characteristics, type of work, and ownership of selected goods were collected, and principal component analysis (PCA) was used to classify respondents' SES into a 5-quintile asset index. During

**Table 2. Prevalence of anemia and selected anemia risk factors in women of reproductive age (WRA).**

| Country | Original Survey N | Analysis N | Anemia | Inflammation | Iron Deficiency | Vitamin A Insufficiency | Vitamin B12 Deficiency | Folate Deficiency | Malaria |
|---|---|---|---|---|---|---|---|---|---|
| | | | | | *Malarial* | | | | |
| Côte d'Ivoire | 834 | 388 | 49.2% | 30.3% | 24.6% | 16.1% | 18.2% | 86.2% | 6.1% |
| Cameroon | 731 | 323 | 34.9% | 17.9% | 18.4% | 17.7% | 15.1% | 15.9% | 13.0% |
| Liberia | 1942 | 1875 | 33.5% | 18.5% | 29.2% | 20.5% | | | 17.9% |
| Malawi | 776 | 740 | 21.7% | 13.4% | 15.0% | 18.5% | 12.7% | 11.0% | 16.8% |
| | | | | | *Non-malarial high inflammation* | | | | |
| Azerbaijan | 2643 | 1250 | 39.5% | 35.9% | 42.9% | 7.2% | 18.2% | 3.2% | |
| Cambodia | 705 | 429 | 44.4% | 39.1% | 3.3% | 8.9% | 1.0% | 18.2% | |
| Pakistan | 5988 | 5427 | 50.6% | 31.5% | 43.7% | 63.4% | 51.5% | 53.1% | |
| | | | | | *Medium-to-low inflammation* | | | | |
| Afghanistan | 1043 | 944 | 39.5% | 20.4% | 30.7% | 40.8% | | | |
| Bangladesh | 876 | 825 | 26.9% | 16.8% | 9.6% | 41.2% | 6.9% | 38.3% | |
| Laos | 816 | 812 | 36.0% | 13.9% | 26.3% | | | | |
| Nepal | 2330 | 2128 | 20.5% | 8.8% | 18.6% | 20.5% | | 4.5% | 0.0% |

Anemia defined as hemoglobin adjusted for altitude and smoking <12.0 g/dL, inflammation defined as either CRP >5 mg/L or AGP >1 g/dL, iron deficiency defined as inflammation-adjusted ferritin <15 μg/L, vitamin A insufficiency: retinol binding protein or retinol <1.05 μmol/L, folate deficiency: red blood cell folate <226.5 nmol/L or serum folate <10 nmol/L for radioimmunoassay (<6.8 nmol/L for microbiological assay), vitamin B12 deficiency: serum or plasma B12 <150 pmol/L. Inflammation prevalence of >30% and ≤30% were used to define high inflammation and medium-to-low inflammation.

the data harmonization process, SES was further categorized into three levels with the poorest two quintiles defined as the lowest, third and fourth quintiles as the medium, and the highest quintile as the highest.

Results were stratified into 3 categories: malarial, non-malarial high inflammation burden (>30% inflammation), non-malarial medium/low inflammation burden based on availability of malaria data. Statistical analysis was performed using SAS 9.4 (SAS Institute Inc., Cary, NC, USA) and R 3.4.0 (R Foundation for Statistical Computing, Vienna, Austria).

## Results

Table 1 summarizes the prevalence of anemia and the selected risk factors among 10 national surveys in PSC, grouped by malaria and inflammation burden. Anemia was present in 19–71% of children, depending on the survey. Approximately 24–68% of children were found to have any inflammation, among whom 95% had elevated AGP and 44% had elevated CRP. Iron deficiency and VAD were found in 5–51% and ≤10% (except 39% in Afghanistan) of children, respectively. Among the 4 malarial countries, 27–29% of children tested positive. Blood disorders were present in 47%, 73%, and 8% of children in Malawi, Laos, and Nepal, respectively.

Table 2 summarizes the prevalence of anemia and anemic risk factors across 11 national surveys in WRA, grouped by malaria and inflammation burden. Anemia was found in 21–51% of women. Approximately 9–39% were found to have inflammation, among whom 74% had elevated AGP and 47% elevated CRP. Iron deficiency and vitamin A insufficiency were found in 3–44% and 7–63% of women, respectively. A total of 1–52% and 3–86% of women were found to be vitamin B12 and folate deficient, respectively. Among the 4 malarial countries, 6–18% of women tested positive.

Fig 1 shows the estimated AF of anemia associated with iron deficiency in PSC and WRA, respectively. Overall, AF based on unadjusted and adjusted OR resulted in the highest estimates. Among WRA, AF based on different approaches of PR estimation were generally similar and comparable with average AF regardless of malaria and inflammation burden (10–30% in malarial countries, 3–47% in high inflammation countries, 12–25% in medium-to-low inflammation countries). Among PSC, AF using PR estimation and average AF were 2–21% in malarial countries, 4–23% in high inflammation countries, 6–39% in medium-to-low inflammation countries. In Bangladesh, average AF estimates appeared to be approximately 5% higher compared to Levin's approach. For the remaining countries, adjusted average AF produced comparable estimates with Levin's approach.

Fig 2 shows the estimated AF of anemia associated with inflammation in PSC and WRA, respectively. Disregarding the approaches using OR, adjusted AF were 6–24% and 2–11% in malarial countries, -1-20% and -4–4% in high inflammation countries, 7–17% and 0–6% in medium-to-low inflammation countries among PSC and WRA, respectively. A negative AF in Cambodia was found due to estimated PR being less than 1 (though not statistically significant). In Cameroon and Malawi PSC, AF with unadjusted PR yielded substantially higher estimates than adjusted PR (30% vs. 19% and 27% vs. 7%, respectively). Additionally, in Laos PSC, before adjusting for age, sex, and SES, around 3–6% of anemia was attributable to inflammation but reduced to near 0% after adjustment, indicating the importance of covariate adjustment in AF quantification.

Fig 3 shows the estimated AF of anemia related to malaria in PSC and WRA, respectively. Using adjusted PR, malaria was associated with 2–35% and 2–16% of children and women with anemia across Cameroon, Cote d'Ivoire, Liberia, and Malawi, respectively. AF using adjusted PR yielded slightly higher estimates than unadjusted PR in Malawi (35% vs. 30% in

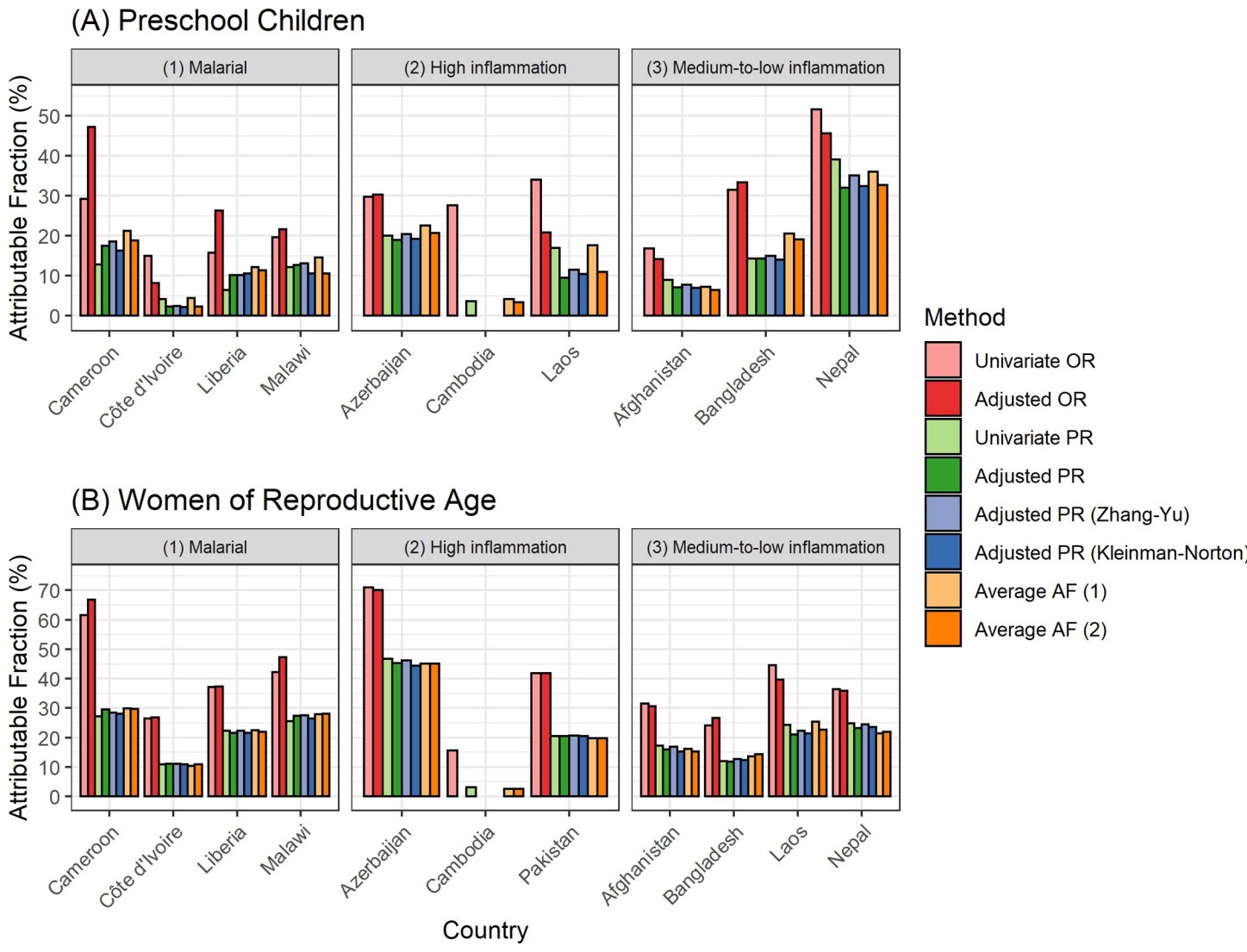

**Fig 1.** Estimated attributable fractions of anemia associated with iron deficiency in (A) preschool children and (B) women of reproductive age. Anemia defined as hemoglobin adjusted for altitude <11.0 g/dL for children and hemoglobin adjusted for altitude and smoking <12.0 g/dL for women. Inflammation-adjusted (using BRINDA regression adjustment method) ferritin concentrations <12 μg/L and <15 μg/L were used to define iron deficiency (ID) in children and women, respectively. Average AF (attributable fractions) (1) considers all exposure variables including inflammation, malaria, blood disorder, and other micronutrient deficiencies depending on data availability. Average AF (2) considers all exposure variables as above and further adjusts for age, sex (for children), and SES. Covariates for adjusted OR and adjusted PR included all available exposure variables, age, sex (for children), and SES. Adjusted AF were not available in Cambodia due to low prevalence of iron deficiency.

PSC and 16% vs. 9% in WRA). In Malawi, average AF generally yielded lower estimates than those using PR with covariate adjustment (11% vs. 13–16% in WRA).

Fig 4 displays the estimated AF of anemia relevant to blood disorders in PSC from three countries. AF estimates using PR ranged 6–14% in Malawi, 25–31% in Cambodia, and 2–4% in Nepal. Again, average AF values were similar. The results of other exposure variables, including VAD, B12 deficiency, and folate deficiency for PSC and WRA are presented in S1 and S2 Tables. Overall, there were no appreciable differences in the AF estimates across different adjusted PR approaches.

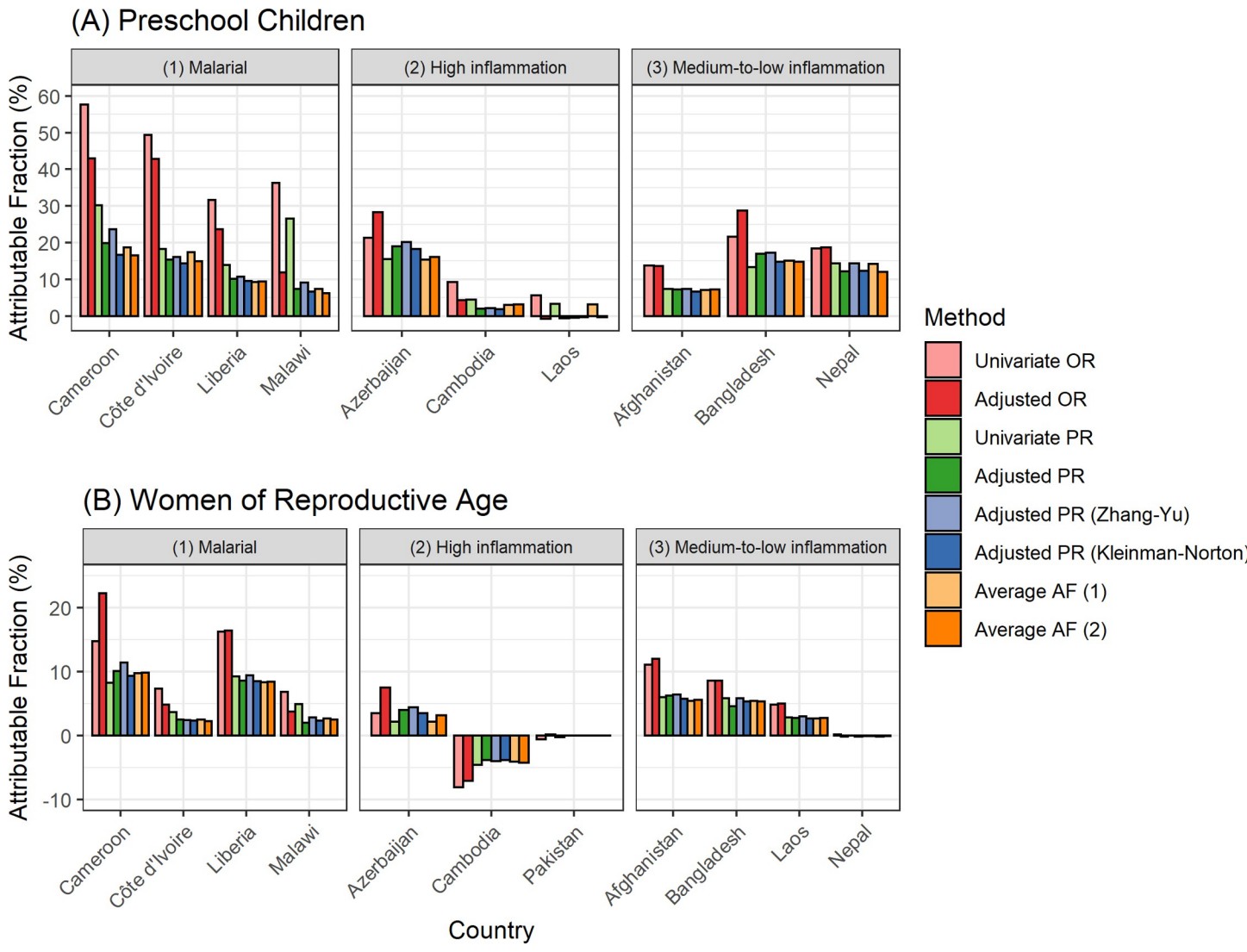

**Fig 2.** Estimated attributable fractions of anemia associated with inflammation in (A) preschool children and (B) women of reproductive age. Anemia defined as hemoglobin adjusted for altitude <11.0 g/dL for children and hemoglobin adjusted for altitude and smoking <12.0 g/dL for women. Inflammation defined as either CRP >5 mg/L or AGP >1 g/dL. Average AF (attributable fractions) (1) considers all exposure variables including malaria, blood disorder, and micronutrient deficiencies depending on data availability. Average AF (2) considers all exposure variables and adjusts for age, sex (for children), and SES. Covariates for adjusted OR and adjusted PR included all available exposure variables, age, sex (for children), and SES.

## Discussion

This study illustrates the use of different approaches to quantify the contribution of multiple known anemia risk factors in PSC and WRA using data from high quality national nutrition cross sectional surveys. AF using adjusted PR from three different approaches yielded nearly identical estimates. Using unadjusted and adjusted OR yielded the highest estimates, which are likely biased given that anemia was a common outcome in PSC and WRA among the study countries. Although the interpretation of average AF involves reduction of cases after removal of other risk factors, which is different from traditional AF, average AF estimates appeared to be comparable to those using adjusted PR in our applications; the small differences observed are unlikely to affect programmatic decisions. These findings are timely as population-based evaluations of anemia are frequently conducted [2, 5, 6, 21, 38, 43–45] and require guidance

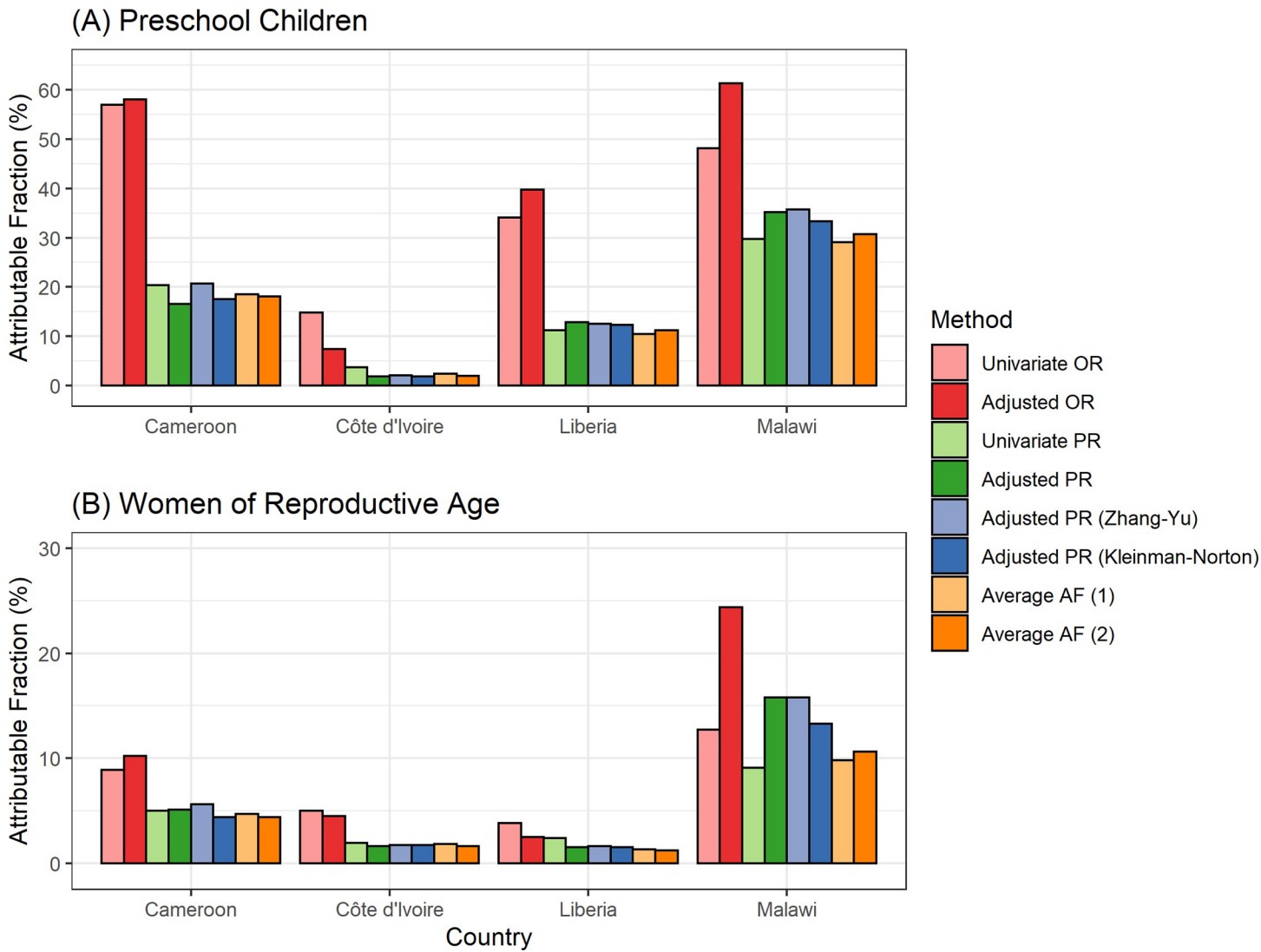

**Fig 3.** Estimated attributable fractions of anemia associated with malaria in (A) preschool children and (B) women of reproductive age. Anemia defined as hemoglobin adjusted for altitude <11.0 g/dL for children and hemoglobin adjusted for altitude and smoking <12.0 g/dL for women. Average AF (attributable fractions) (1) considers all exposure variables including malaria, blood disorder, and micronutrient deficiencies depending on data availability. Average AF (2) considers all exposure variables and adjusts for age, sex (for children), and SES. Covariates for adjusted OR and adjusted PR included all available exposure variables, age, sex (for children), and SES.

on best practice for these types of analyses to inform public health interventions to combat anemia [4].

Disregarding the AF estimates using OR, the results revealed an overall AF in the range of 2–52% for children and 3–71% for women related to iron deficiency, depending on the prevalence of iron deficiency and its association with anemia. Although our goal was not to derive nationally representative AF of anemia, the relative contributions of micronutrient deficiencies, inflammation, and malaria are generally consistent with the findings from other AF of anemia studies [5, 8].

In our application, unadjusted AFs may or may not yield different AF estimates compared with adjusted ones. When an exposure factor is correlated with another one or when the association with anemia is confounded by other distal demographic factors (e.g., SES), univariate analysis could lead to an overestimation or underestimation of AF. Hence, crude estimation of

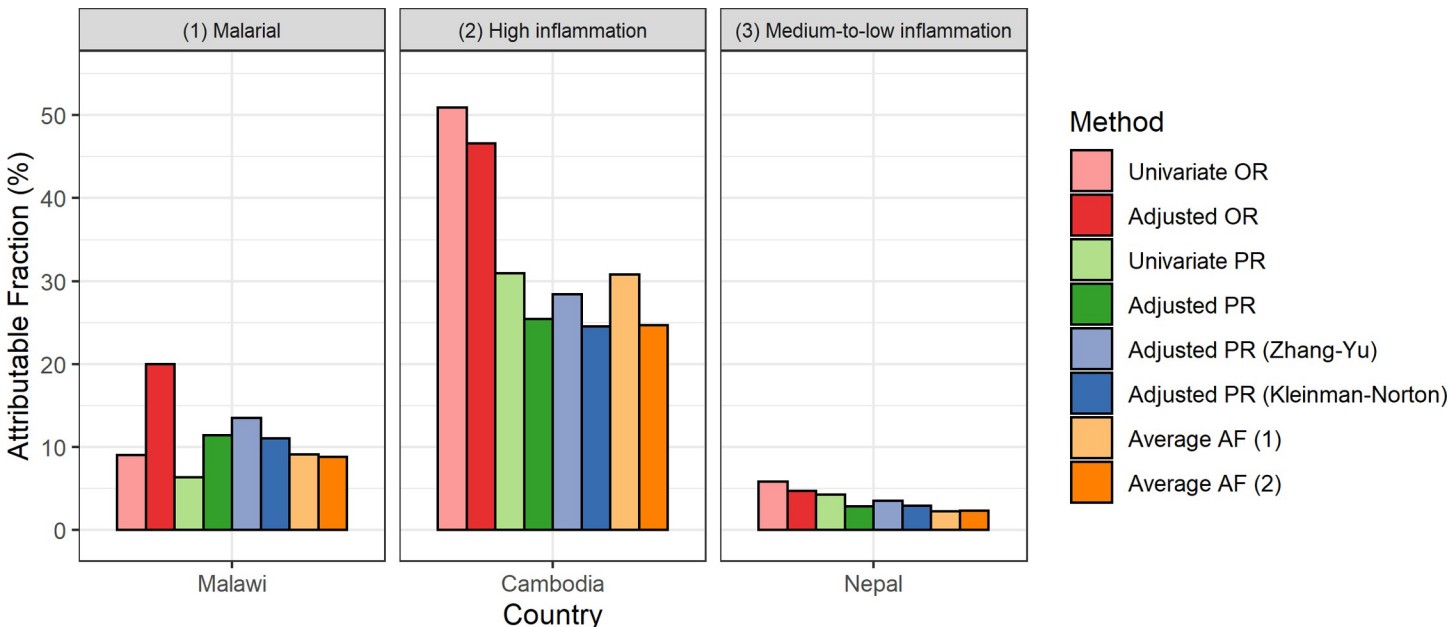

**Fig 4. Estimated attributable fractions of anemia associated with blood disorder in preschool children.** Anemia defined as hemoglobin adjusted for altitude <11.0 g/dL. Average AF (attributable fractions) (1) considers all exposure variables including malaria, blood disorder, and micronutrient deficiencies depending on data availability. Average AF (2) considers all exposure variables and adjusts for age, sex (for children), and SES. Covariates for adjusted OR and adjusted PR included all available exposure variables, age, sex (for children), and SES.

AF should be abandoned, and any confounding should be minimized in cross-sectional or cohort studies. It is important to develop an appropriate and sensible regression model to estimate PR, given that the subsequent AF estimate can change substantially depending on covariates and their functional forms in the model.

We used OR to calculate AF for illustration purposes and comparisons with other approaches to show that using OR in an AF calculation may produce incorrect results. It has been suggested that OR should not be used to approximate RR or PR when the disease is not rare (<10%), since OR always overstates RR or PR for common diseases [39, 46]. The prevalence of anemia among these study countries was estimated to be 19–71% in children and 20–50% in women, and recent estimates of the global anemia prevalence shows that anemia prevalence exceeds 10% in all geographic regions [3]. Thus, the AF estimates using the Levin's formula with unadjusted or adjusted OR from logistic regression are clearly biased, and would not usable in most countries.

The average AF method may be applicable for solving the problem of correlated attributions for the prevalence of a disease in the population. Among the exposure variables being investigated, the sum of average AF does not exceed 100% [18]. On the other hand, Levin's formula assumes that each risk factor is the first to be eliminated. Individual AF estimates can sum to greater than 100%, resulting in a counter-intuitive scenario where more than 100% of cases are preventable. It has been argued that a sum of more than 100% is possible due to shared etiologic responsibility among exposures [47], as anemia can be prevented in multiple different ways [48]. As such, it is incorrect to sum up AF from different risk factors when average AF is not used. Comparisons of AF for different risk factors alone can be useful for policy makers.

A recent paper proposed to estimate sequential and average AF based on simulation from causal Bayesian networks by modeling exposure-exposure and exposure-disease interrelationships [49]. Strong assumptions of a directed acyclic graph and component probability

distributions are required to achieve consistent estimations. We attempted to apply this approach to the BRINDA data under the assumption of a number of biologically plausible causal structures but found AF estimates of iron deficiency to be negligible. One reason may be our overly-simplistic model by using dichotomous/categorical variables along with the presence of unmeasurable confounders. Additionally, establishing a causal framework and estimating these relationships in a cross-sectional dataset is prone to problems due to unknown directions of causal relationships.

Our study has limitations. First, cross-sectional data were analyzed, so we were not able to make true causal inference on "risk" reduction of anemia. In other words, there is potential for reverse causality. For example, infections can lead to anemia, but it is possible that having anemia makes one more vulnerable to infectious diseases. Moreover, the interpretations of PR and RR are different. AF in a cross-sectional study represents the excess of *prevalent* cases of anemia that can be attributed to the exposure of interest [16]. There is not a gold standard approach to use to compare different AF estimation methods. Second, limited data were available on other known risk factors for anemia (e.g., infections such as human immunodeficiency virus, hookworm and other soil-transmitted helminth; renal disease; blood loss) that could be considered as covariates. Third, hemoglobin concentrations measured by HemoCue 201+ and 301 are not fully comparable, which may contribute to uncertainty around the definition of anemia. However, the slight discrepancy between the two hemoglobinometer models has minimal impact on anemia prevalence [50]. Last, as with all methods, AF estimates may be biased due to missing important covariates or misspecifications of covariate functional forms. Our current models did not consider continuous or other categorizations than binary exposure variables. Calculations of adjusted AF using continuous variables are certainly feasible, but an AF corresponding to a continuous exposure variable might not be straightforward to interpret for program or policy decisions. Nevertheless, we do not consider any of these limitations specific to comparing AF methods.

The USAID Advancing Nutrition Anemia Task Force [4] has emphasized the need for interpreting evidence on the nutritional, physiological, and genetic aspects of anemia, and for developing approaches that start with context-specific assessment of anemia and its risk factors. AF may be one metric to help prioritize interventions to reduce anemia prevalence although the potential limitations of AF values derived from cross-sectional data need to be considered. Our study demonstrated that AF quantification can be obtained in different ways with Poisson regression being the most straightforward approach. The similarity across methods provides researchers flexibility in selecting AF approaches.

## Supporting information

**S1 Table. Estimated attributable fractions of anemia associated with various risk factors in preschool children.**
(DOCX)

**S2 Table. Estimated attributable fractions of anemia associated with various risk factors in women of reproductive age.**
(DOCX)

## Acknowledgments

We acknowledge the contributions of the BRINDA working group and steering committee members (https://brinda-nutrition.org).

The findings and conclusions in this report are those of the authors and do not necessarily represent the official position of the US Centers for Disease Control and Prevention.

## Author Contributions

**Conceptualization:** Yi-An Ko, Janet M. Peerson, Parminder S. Suchdev.

**Data curation:** Yi-An Ko, Hanqi Luo.

**Formal analysis:** Yi-An Ko, Anne M. Williams, Janet M. Peerson.

**Funding acquisition:** Melissa F. Young, Parminder S. Suchdev.

**Investigation:** Yi-An Ko, Janet M. Peerson.

**Methodology:** Yi-An Ko, Janet M. Peerson.

**Project administration:** Melissa F. Young, Parminder S. Suchdev.

**Resources:** Melissa F. Young, Parminder S. Suchdev.

**Software:** Yi-An Ko.

**Supervision:** Parminder S. Suchdev.

**Validation:** Yi-An Ko.

**Visualization:** Yi-An Ko.

**Writing – original draft:** Yi-An Ko, Parminder S. Suchdev.

**Writing – review & editing:** Yi-An Ko, Anne M. Williams, Janet M. Peerson, Hanqi Luo, Rafael Flores-Ayala, James P. Wirth, Reina Engle-Stone, Melissa F. Young, Parminder S. Suchdev.

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
