## [Decision Letter · Decision Letter 0]

24 Aug 2022

Approaches to quantify the contribution of multiple anemia risk factors in children and women from cross-sectional national surveys

PGPH-D-22-00788

Dear Yi An Ko 

We are pleased to inform you that your manuscript 'Approaches to quantify the contribution of multiple anemia risk factors in children and women from cross-sectional national surveys' has been provisionally accepted for publication in PLOS Global Public Health.

Best regards,

Marianella Herrera-Cuenca, MD, PhD

Academic Editor

Reviewer Comments (if any, and for reference):

Reviewer's Responses to Questions

**Comments to the Author**

1. Does this manuscript meet PLOS Global Public Health’s publication criteria? Is the manuscript technically sound, and do the data support the conclusions? The manuscript must describe methodologically and ethically rigorous research with conclusions that are appropriately drawn based on the data presented.

Reviewer #1: Yes

2. Has the statistical analysis been performed appropriately and rigorously?

Reviewer #1: Yes

3. Have the authors made all data underlying the findings in their manuscript fully available (please refer to the Data Availability Statement at the start of the manuscript PDF file)?

Reviewer #1: Yes

4. Is the manuscript presented in an intelligible fashion and written in standard English?

Reviewer #1: Yes

5. Review Comments to the Author

Reviewer #1: Anemia is a serious problem of public health with negative consequences on growth, development an global health in children, associated in many cases with evident malnutrition and/or micronutrient , specially in groups at high risk: children and women at reproductive ages.

The attributable fraction in a population is an epidemiologic measure used to quantify the proportional reduction in population risk of disease or mortality, while distributions of other risk factors remain unchanged in the population.

Excellent article in everyone of its sections: aim, methods, results including statistical analysis.

Publication recommended

6. PLOS authors have the option to publish the peer review history of their article (what does this mean?). If published, this will include your full peer review and any attached files.

**Do you want your identity to be public for this peer review?** For information about this choice, including consent withdrawal, please see our Privacy Policy.

Reviewer #1: No
